# Integration of AIEgens into covalent organic frameworks for pyroptosis and ferroptosis primed cancer immunotherapy

Liang Zhang[1,2,3,5], An Song[1,5], Qi-Chao Yang[1], Shu-Jin Li[1], Shuo Wang[1], Shu-Cheng Wan[1], Jianwei Sun[2], Ryan T. K. Kwok[2], Jacky W. Y. Lam[2] ✉, Hexiang Deng[3] ✉, Ben Zhong Tang[2,4] ✉ & Zhi-Jun Sun[1] ✉

Immunogenic programmed cell death, such as pyroptosis and ferroptosis, efficiently induces an acute inflammatory response and boosts antitumor immunity. However, the exploration of dual-inducers, particularly nonmetallic inducers, capable of triggering both pyroptosis and ferroptosis remains limited. Here we show the construction of a covalent organic framework (COF-919) from planar and twisted AIEgen-based motifs as a dual-inducer of pyroptosis and ferroptosis for efficient antitumor immunity. Mechanistic studies reveal that COF-919 displays stronger near-infrared light absorption, lower band energy, and longer lifetime to favor the generation of reactive oxygen species (ROS) and photothermal conversion, triggering pyroptosis. Because of its good ROS production capability, it upregulates intracellular lipid peroxidation, leading to glutathione depletion, low expression of glutathione peroxidase 4, and induction of ferroptosis. Additionally, the induction of pyroptosis and ferroptosis by COF-919 effectively inhibits tumor metastasis and recurrence, resulting in over 90% tumor growth inhibition and cure rates exceeding 80%.

Phototherapy has emerged as a promising oncological intervention option owing to its exceptional intrinsic characteristics, such as spatial accuracy, non-invasiveness, and minimal side effects[1–6]. Apoptosis has been demonstrated to be the major mode of cell death after phototherapy. However, its therapeutic effect is usually limited due to the apoptosis resistance[7–9]. Recently, some immunogenic programmed cell death modes, such as pyroptosis and ferroptosis, have been proven to efficiently induce an acute inflammatory response to produce robust antitumor immune activity, which is promising for cancer immunotherapy[10–14]. Efforts have been made to explore metal-containing pyroptosis and ferroptosis inducers, such as $Fe^{3+}$ phenolics, Bi–Au nanoclusters, GOx–Mn, $LaFeO_3$, CaNMs, and metal–organic frameworks[15–22]. Unfortunately, in all of these studies, the introduction of metal species may trigger undesirable toxicity and potentially detrimental effects. Thus, it is an appealing yet seriously challenging task to construct non-metallic pyroptosis and ferroptosis inducers, especially dual-inducers.

In this contribution, a non-metallic covalent organic framework (COF) based pyroptosis and ferroptosis dual-inducer was rationally designed by integrating aggregation-induced emission luminogens

[1]State Key Laboratory of Oral & Maxillofacial Reconstruction and Regeneration, Key Laboratory of Oral Biomedicine Ministry of Education, Hubei Key Laboratory of Stomatology, School & Hospital of Stomatology, Wuhan University, Wuhan 430079, China. [2]Department of Chemistry, and The Hong Kong Branch of Chinese National Engineering Research Center for Tissue Restoration and Reconstruction, The Hong Kong University of Science and Technology, Clear Water Bay, Kowloon, Hong Kong 999077, China. [3]Key Laboratory of Biomedical Polymers-Ministry of Education, College of Chemistry and Molecular Sciences, Wuhan University, Luojiashan, Wuhan 430072, China. [4]Shenzhen Institute of Aggregate Science and Technology, School of Science and Engineering, The Chinese University of Hong Kong, Shenzhen, Guangdong 518172, China. [5]These authors contributed equally: Liang Zhang, An Song. ✉e-mail: chjacky@ust.hk; hdeng@whu.edu.cn; tangbenz@cuhk.edu.cn; sunzj@whu.edu.cn

(AIEgens) into COF skeletons to elicit a strong inflammatory response and boost antitumor immunity. Nanoscale COFs are crystalline organic porous materials and have attracted much attention in the field of cancer phototherapy owing to their excellent light absorption, high porosity, photostability, and good biocompatibility[23–27]. Currently, the major cell death mode caused by COF-mediated phototherapy is apoptosis. However, others and we showed the feasibility of decorating metal-containing inducers onto the skeleton or into the pores of COFs to trigger pyroptosis or ferroptosis[28–30]. However, it remains challenging to create non-metallic COF-based pyroptosis or ferroptosis inducers due to the stringent requirements of ROS generation efficiency. Traditional COF-based photosensitizers (PSs) suffer from strong undesirable aggregation-caused quenching (ACQ) effects. This leads to no or weak emission and limited capability to generate reactive oxygen species (ROS), ultimately failing to trigger pyroptosis or ferroptosis. In this work, a pyroptosis and ferroptosis dual-inducer, called COF-919, was rationally designed by integrating planar and twisted motifs with AIE characteristics[31,32] into the COF skeleton. Such a kind of planar plus twisted AIE COF has several advantages over the existing COF-based photosensitizers. First, in comparison to traditional COFs with all planar or twisted motifs (e.g., COF-909 and COF-818), this planar plus twisted AIE COF (COF-919) displayed better near-infrared (NIR) light absorption, lower band energy and a longer lifetime to contribute good phototherapeutic effect, which is favorable for inducing pyroptosis. Second, benefiting from its good ROS production capability, intracellular lipid peroxidation (LPO) is also upregulated, which leads to glutathione depletion, low glutathione peroxidase 4 (GPX4) expression, and the induction of a GPX4-related ferroptosis process. Third, the superb photothermal therapy (PTT) performance of COF-919 helps improve the pyroptosis and ferroptosis process. Finally, the high porosity of AIE COFs not only effectively avoids the undesirable ACQ and poor photosensitization effects but also facilitates the ROS diffusion process[33,34]. These advantages make COF-919 favorable for inducing pyroptosis and ferroptosis simultaneously (Fig. 1).

In this work, COF-919-mediated photodynamic and photothermal combination therapy elicits GPX4-related ferroptosis and gasdermin E (GSDME)-dependent pyroptosis, inducing an acute inflammatory response and boosting cancer immunotherapy. Specifically, GSDME is cleaved by overexpressed cleaved Caspase-3 to produce N-terminal domains, which further assemble into transmembrane pores and swell

with large bubbles. During this process, the release of abundant cell contents, such as high mobility group protein B1 (HMGB1), lactic dehydrogenase (LDH), and adenosine triphosphate (ATP), induces a strong immune response. On the other hand, the successful induction of ferroptosis by COF-919 is revealed by the overexpression of representative pyroptosis indicators, a decreased GSH/GSSG ratio, an increased cellular iron level, and a shriveled mitochondrial morphology after treatment with COF-919. As a result, COF-919 induces pyroptosis and ferroptosis, promoting the proportion of T cell infiltration in the microenvironment while reducing the proportion of regulatory T cells (Tregs) and immunosuppressive myeloid-derived suppressor cells (MDSCs) in the macroenvironment. This indicates the significant potential of COF-919 in the induction of pyroptosis and ferroptosis. Importantly, COF-919 treatment synergistically improves the response rate of immune checkpoint blockade therapy by αPD-1, and the combination of COF-919 + αPD-1 demonstrates a favorable synergistic effect in inhibiting tumor metastasis and recurrence, resulting in over 90% tumor growth inhibition and cure rates exceeding 80%.

## Results

### Preparation and characterization of COF-818 and COF-919

COF-818 was constructed from two twisted monomers, namely 5',5'''-bis(4-formylphenyl)-[1,1':3',1'':4'',1''':3''',1''''-quinquephenyl]-4,4''''-dicarbaldehyde (M-TPh) and $N^1,N^{1'}$-(1,4-phenylene)bis($N^1$-(4-aminophenyl)benzene-1,4-diamine) (M-TPA), while COF-919 was constructed from a planar AIE monomer, called 4',4''''-(1,4-phenylene)bis(([2,2':6',2''-terpyridine]-5,5''-dicarbaldehyde)) (M-Tpy) and M-TPA (Figs. 2a to 2c, Supplementary Fig. 1-6). The AIE properties of M-TPh, M-Tpy, and M-TPA were investigated in mixed solutions of tetrahydrofuran (THF) and water ($H_2O$) with different $H_2O$ ratios. When dissolved in THF, both M-TPh and M-TPy were almost nonemissive; by gradually increasing the $H_2O$ ratio, the emission intensity was intensified, with the intensity at 90% water fraction being approximately 8 and 18 times higher than that in pure THF for M-TPh and M-TPy, respectively. This clearly unveiled the AIE properties of these monomers (Fig. 2b, d). In contrast, the emission of M-TPA became weaker in the THF/$H_2O$ mixture with increased water fraction, demonstrating the ACQ property of M-TPA (Supplementary Fig. 7). The accurate structures of COF-818 and COF-919 were determined by powder X-ray diffraction (PXRD) patterns and Pawley refinement. As shown in Fig. 2b, COF-818 exhibited four diffraction peaks at 3.19°, 5.11°, 6.18° and 10.31°, corresponding to the *hlk*

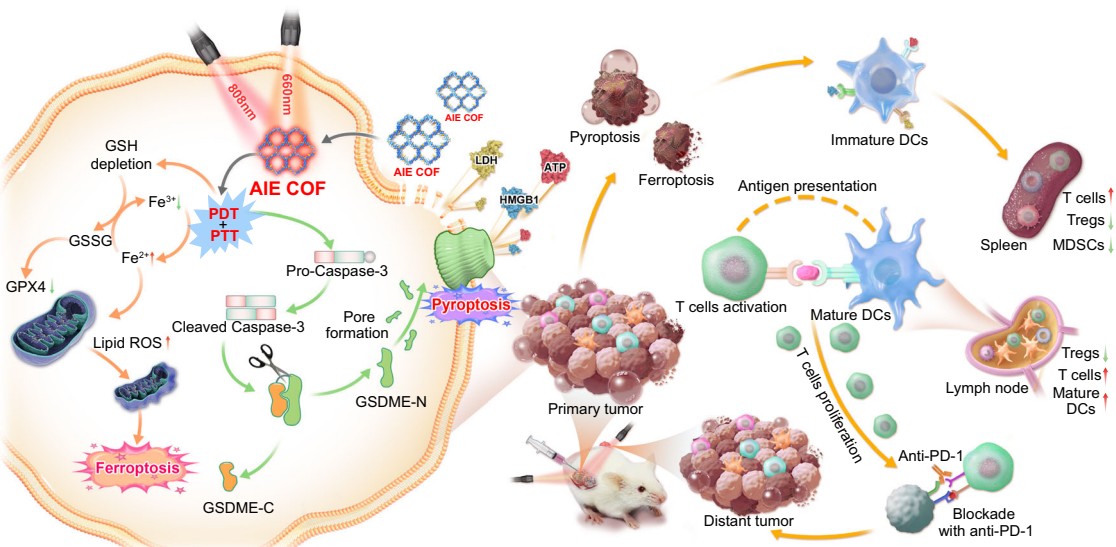

**Fig. 1 | Illustration of the mechanism of COF-919-mediated phototherapy.** Planar and twisted motifs were integrated into covalent organic frameworks for pyroptosis and ferroptosis-primed cancer immunotherapy.

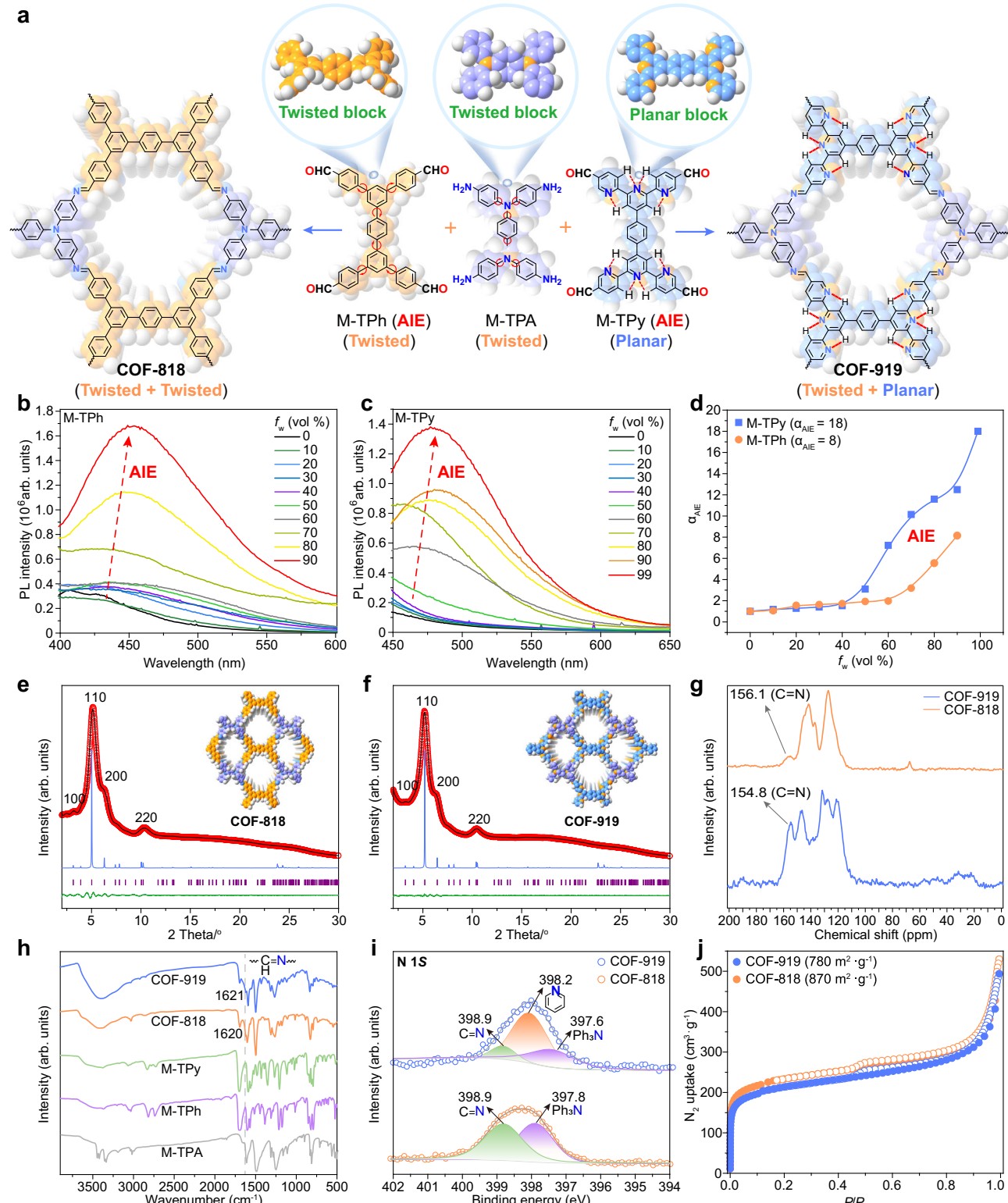

**Fig. 2 | Construction and characterization of AIE COFs. a** Illustration of the construction of COF-818 with two twisted monomers and COF-919 with planar and twisted monomers. PL spectra of M-TPh (**b**) and M-TPy (**c**) in THF/H₂O solutions with different water fractions ($f_w$). **d** Plots of the relative emission intensity ($I/I_0$) of M-TPh and M-TPy versus increased water fraction. PXRD patterns of COF-818 (**e**) and COF-919 (**f**). **g** Solid-state NMR spectra of AIE COFs. **h** IR spectra of COF-818 and COF-919. **i** XPS spectra of COF-818 and COF-919. **j** N₂ adsorption isotherms of COF-818 and COF-919. Source data are provided as a Source Data file.

values of 100, 110, 200, and 220 (Fig. 2e). Similarly, the PXRD patterns of COF-919 displayed four diffraction peaks at 3.15°, 5.18°, 6.21°, and 10.45°, corresponding to the *hlk* values of 100, 110, 200, and 220, respectively (Fig. 2f, Supplementary Tables 1–3). Pawley refinement

was employed to refine the COF structures, which showcased good agreement with their experimental data, with ($R$wp, $R$p) values of (1.11%, 0.85%) for COF-818, and (0.58%, 0.47%) for COF-919, respectively (Fig. 2e, f). The formation of imine linkage in the COF-818 and

COF-919 skeletons was evidenced by the appearance of characteristic solid-state NMR peaks at 156.1 and 154.8 ppm (Fig. 2g); a peak at 398.9 eV in the X-ray photoelectron spectroscopy (XPS) spectra (Fig. 2i); and peaks at 1620 cm$^{-1}$ and 1621 cm$^{-1}$ in the Fourier transform infrared (FT-IR) spectra (Fig. 2h). The permanent porosity of these AIE COFs was proven by N$_2$ uptake experiment, with Brunauer–Emmett–Teller surface areas values of 870 and 780 m$^2$ g$^{-1}$ (Fig. 2j, Supplementary Fig. 8).

## Photophysical properties of COF-818 and COF-919

The phototherapy efficacy of COF-based PSs is dictated by their specific photophysical properties, including light absorption, steady-state fluorescence, lifetime, and band energy (Fig. 3a)[35]. To study these photophysical properties, a series of spectroscopic experiments were carried out. Ultraviolet/visible diffuse reflectance spectroscopy was conducted to investigate the light absorption properties of these AIE COFs. A clear red-shift was observed in the spectrum of COF-919 in comparison with that of COF-818, demonstrating the better coefficient absorption of COF-919 (Fig. 3b, Supplementary Figs. 9–11). Then, both steady-state and time-resolved photoluminescence (PL) spectra were obtained to study the charge-transfer properties of these COFs. As depicted in Fig. 3c, d, compared with COF-818, COF-919 showed decreased steady-state PL and a longer lifetime, which unveiled faster electron-hole separation and a lower charge recombination rate upon photoexcitation, critical for improving the PDT performance (Fig. 3c, d). Furthermore, the band energy properties, including the band gap and valence band, were also studied by ultraviolet photoelectron spectroscopy, where COF-919 possessed a lower valence band and band gap compared to that of COF-818 (Fig. 3e–h). These results, combined with the calculation results of COF-818 and COF-919, where COF-919 also lowers band energy (Fig. 3i), unambiguously reveal the excellent light absorption of COF-919.

## In vitro investigation of the photothermal and photodynamic therapy of COF-818 and COF-919

Considering the excellent photophysical properties of COF-919, we then studied its potential to act as a photosensitizer for photothermal and photodynamic therapy (Fig. 4a). As shown in Fig. 4b, both COF-818 and COF-919 can generate heat under 808 nm laser irradiation, and COF-919 displayed better photothermal conversion capability, owing to its better photophysical properties (Fig. 4c, d). The excellent photothermal conversion of COF-919 was further confirmed by experiments conducted in vitro and in vivo (Fig. 4e, f). Subsequently, the ROS generation ability of these AIE COFs was evaluated using 2,7-dichlorodihydrofluorescein diacetate (DCFH-DA) as a ROS detection probe[36]. The results showed that after adding DCFH-DA to the COF samples, no obvious fluorescence intensity change was observed in either the control or the COF-818 sample. However, a drastic enhancement in fluorescence intensity was observed in the COF-919 sample under the same conditions, demonstrating its excellent ROS generation capability and outlining the superiority of COF-919 for phototherapy (Fig. 4g–i). Then, nanoscale COF samples were successfully generated by ultrasound treatment in PBS. The sample size determined by dynamic light scattering (DLS) and transmission electron microscopy (TEM) was ~200 nm, which is suitable for in vivo experiments. (Fig. 4d). As displayed in Supplementary Fig. 12, after 12 h of coculture, 4T1 cells showed a clear red fluorescence signal, indicating that these nanoscale COFs could be effortlessly engulfed by 4T1 cells. We then evaluated the cytotoxicity of COF-818 and COF-919 by using an MTT assay. After treatment with COF solutions with different concentrations, over 95% of the 4T1 cells survived, demonstrating the good biocompatibility of COF-818 and COF-919. In contrast, after laser irradiation, less than 25% of the 4T1 cells survived after treatment with COF-919 (Supplementary Figs. 13–15), unveiling the higher photo cytotoxicity of COF-919.

## Pyroptosis-inducing capability of COF-818 and COF-919

Based on previous studies, the release of a large amount of ROS caused by PDT may trigger acute inflammation and elicit pyroptosis, which is characterized by the formation of large bubbles on the membrane (Fig. 5a)[37–40]. The gasdermin family proteins have been proven to be crucial for pyroptotic cell death[41,42]. The classical pyroptosis process is usually activated by Caspase-1, where the protein gasdermin D (GSDMD) is cleaved to release GSDMD-N domains and form large bubbles[43–45]. Recently, a gasdermin family protein-dependent pyroptosis process was reported by Shao et al., which can be activated by Caspase-1/3/4/5/11[46–49]. To investigate whether these COFs could trigger pyroptosis, optical confocal microscopy was used to monitor morphological changes in situ. As shown in Fig. 5f, 4T1 cells treated with COF-919 + Laser generated bubbles, and obvious bubbles were formed with 660 nm plus 808 nm laser irradiation, outlining the importance of dual-wavelength laser irradiation. The occurrence of pyroptosis was further confirmed by western blotting. Although both COF-818 and COF-919 could cause cleaved Caspase-3 overexpression, GSDME-N was only overexpressed in COF-919-treated 4T1 cells (Fig. 5e). This revealed that pyroptosis was successfully induced by COF-919. In contrast, COF-818 only triggered apoptosis. This was further evidenced by the quantitative evaluation of the representative pyroptosis parameters, such as ATP, LDH, and HMGB1. A notable increase in ATP and LDH concentrations was found in COF-919-treated 4T1 cells under 660 and/or 808 nm laser irradiation (Fig. 5b, c). In contrast, no distinct change in ATP or LDH concertation was observed in other groups, including phosphate buffer solution (PBS), laser, and COF-818. Similarly, a sharp decrease in the concertation of HMGB1 was detected in only COF-919 treated 4T1 cells under 660 and/or 808 nm laser irradiation (Fig. 5d). These results clearly revealed the good pyroptosis-inducing efficacy of COF-919.

## Ferroptosis-inducing capability of COF-818 and COF-919

Generally, generations of sufficient ROS in lipids will result in intracellular lipid peroxidation, GSH depletion, and low expression of GPX4 to trigger another immunogenic programmed cell death mode, called ferroptosis, and PTT can facilitate ferroptosis efficacy[50–52]. We hypothesized that the excellent PDT and PTT performance of COF-919 may elicit ferroptosis (Fig. 6a). To test our hypothesis, the expression of three representative ferroptosis indicators, including GPX4, transferrin receptor (TFRC), solute carrier family 7 and member 11 (xCT), was assessed by western blot. As seen in Fig. 6b, Supplementary Fig. 16, a clear reduction in the expression of GPX4 and xCT was observed in 4T1 cells treated with COF-919 + 660 nm, COF-919 + 808 nm, and COF-919 + 660 + 808 nm groups. Among them, COF-919 + 660 + 808 nm treatment resulted in the lowest expression, outlining the importance of PDT/PTT combination therapy. Similarly, TFRC, another ferroptosis indicator, was also found to be obviously overexpressed after treatment with COF-919 + 660 + 808 nm. These results, combined with the drastic enhancement in cellular iron levels and decreased GSH/GSSG ratio, indicated the occurrence of strong ferroptosis caused by COF-919 + 660 + 808 nm (Fig. 6c, d). The high ferroptosis-inducing performance of COF-919 was further confirmed by detecting the lipid ROS levels, where the 4T1 cells treated with COF-919 + 660 + 808 nm displayed the strongest green fluorescence (Fig. 6e). Moreover, the TEM results also indicated that compared with normal ones, the morphology mitochondria treated with that of COF-919 + 660 + 808 nm was notably shriveled (Fig. 6f). These results clearly demonstrated the successful elicitation of ferroptosis by COF-919-mediated PDT and PTT.

## In vivo therapeutic effect of AIE COFs

To in vivo validate the therapeutic effect of COF-818 and COF-919, a 4T1 tumor-bearing mouse model was set up (Fig. 7a). As shown in Fig. 7b–d, the growth rate of the 4T1 tumors was effectively inhibited

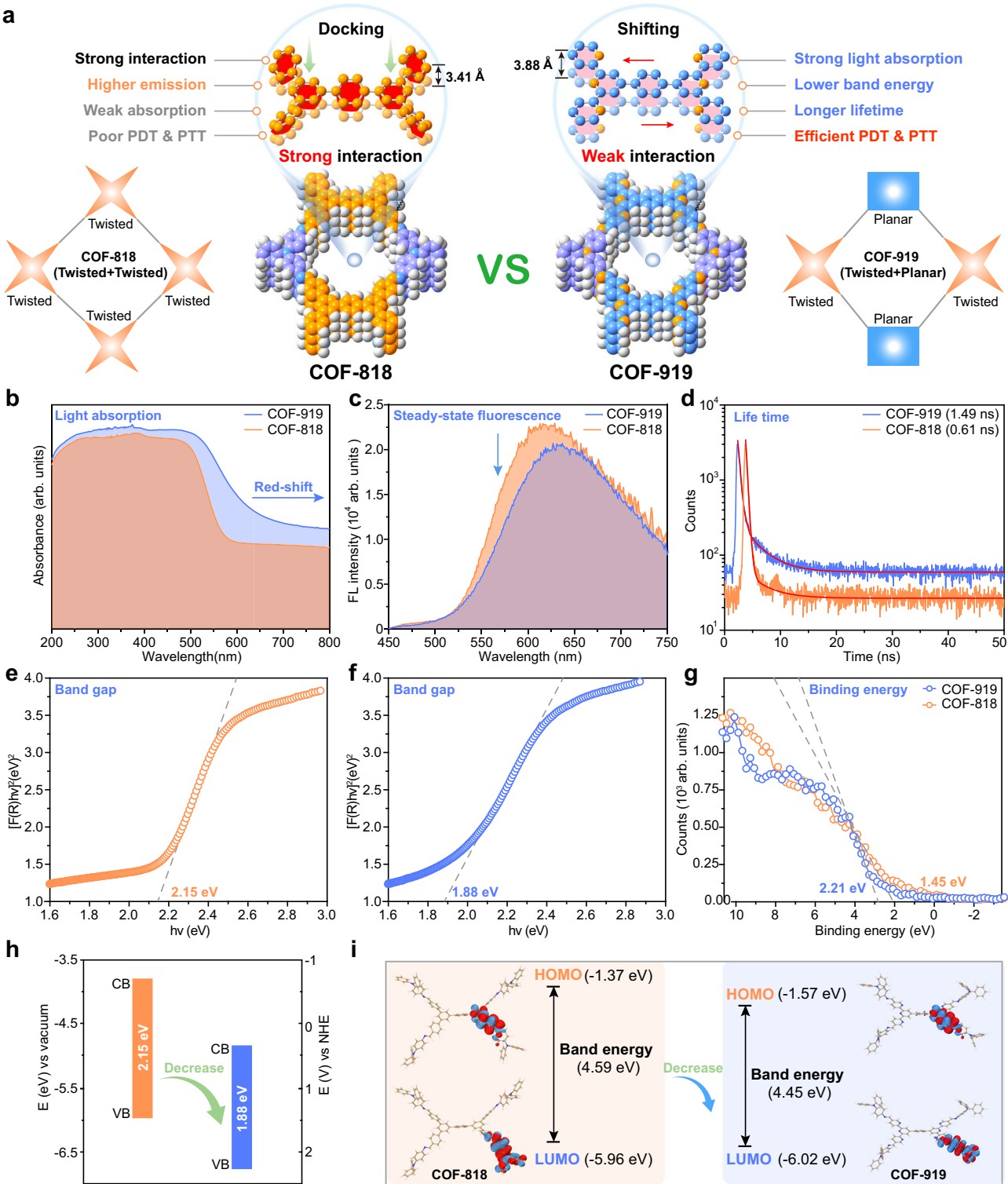

**Fig. 3 | Characterization of the photophysical properties of AIE COFs.**
**a** Illustration of the advantages of COF-919 with planar and twisted monomers.
**b** UV/vis light absorption spectra of AIE COFs. Steady-state (**c**) and time-resolved (**d**)
photoluminescence spectra of AIE COFs. **e, f** Band gap of COF-818 and COF-919. **g** Binding energy of COF-818 and COF-919. **h** Band structure of COF-818 and COF-919. **i** DFT calculations for COF-818 and COF-919. Source data are provided as a Source Data file.

after treatment with COF-919 irradiated under 660 and/or 808 nm laser and almost all of the tumors were eliminated after treated with COF-919 under 660 and 808 nm dual-wavelength laser irradiation. This indicated the excellent therapeutic effect of COF-919 + 660 + 808 nm. Then, western blotting and immunohistochemical staining were

performed to analyze tumor tissues from each group of animals after euthanasia. The Hematoxylin and eosin (H&E) staining images of the major organs showed no apparent histopathological abnormalities after different treatments (Supplementary Fig. 17). This, combined with no obvious changes in body weight (Fig. 7c), demonstrated the

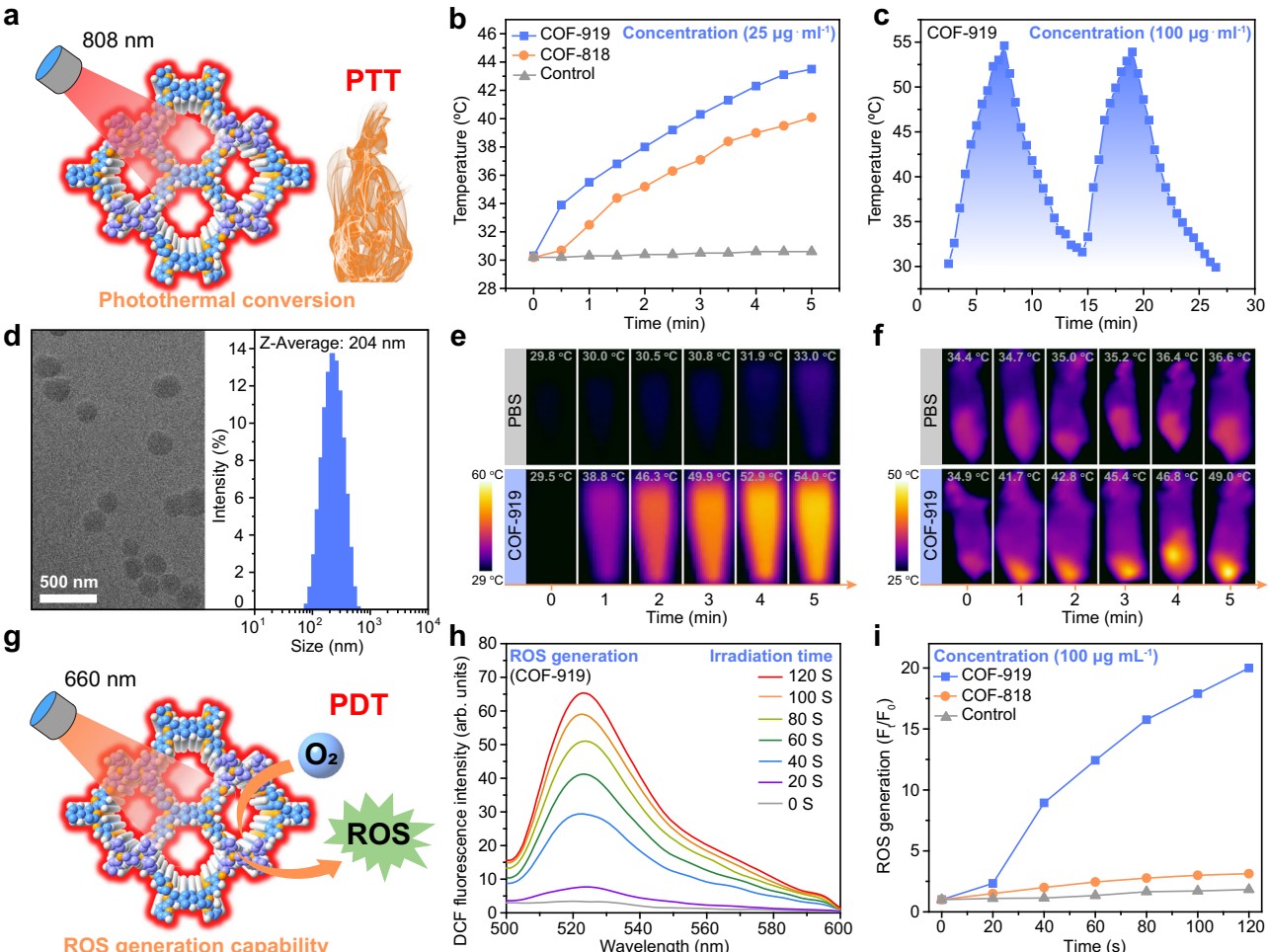

**Fig. 4 | Characterization of the phototherapy capability of COF-818 and COF-919. a** Illustration of the AIE COF-based photothermal conversion. **b** Temperature changes in the control, COF-818 and COF-919 samples under the same conditions. **c** Photothermal stability of COF-919 during two heating-cooling cycles. **d** TEM images and DLS of COF-919; scale bar = 500 nm; data were repeated thrice independently with similar results. In vitro (**e**) and in vivo (**f**) thermal images of the control and COF-919 after different irradiation times. **g** Illustration of ROS generated by COF-based photosensitizers Illustration of efficient ROS generated by COF-based photosensitizers. **h**, **i** ROS generation assay. Source data are provided as a Source Data file.

good biocompatibility of these COFs. The immunohistochemical staining results showed a sharp decrease in the proliferation marker Ki67 after treatment with COF-919 + 660 + 808 nm, indicating successful tumor growth inhibition (Fig. 7e, Supplementary Fig. 18). Moreover, after treatment with COF-919 + 660 + 808 nm, granzyme B and CD8 were also overexpressed, demonstrating the clear activation of the immune system (Supplementary Fig. 19). In addition, the cleaved Caspase-3 and 4-Hydroxynonenal (4-HNE) were also overexpressed after treatment with COF-919 + 660 + 808 nm (Fig. 7f, Supplementary Fig. 18). These results, combined with the overexpression of the protein GSDME-N and reduced expression of the GPX4 determined by western blotting after treatment with COF-919 + 660 + 808 nm (Fig. 7g), unveiled the successful occurrence of pyroptosis and ferroptosis within 4T1 tumor tissues.

As immunogenic programmed cell death modes, pyroptosis and ferroptosis have demonstrated their potential to induce an acute inflammatory response to produce antitumor immune activity and boost cancer immunotherapy (Fig. 7h)[53–55]. To investigate the influence of COF-919-mediated pyroptosis and ferroptosis on the immune cell response, a flow cytometry system was utilized to analyze both the draining lymph (DLN) and the spleen of each tumor-bearing mouse (Supplementary Figs. 20–23). The population of dendritic cells (DCs), a kind of first-line cell that initiates adaptive immune responses[56,57], was found to be notably increased in DLN, implying the potential of activated immune cells (Fig. 7m). Second, the proportions of CD4+ and CD8+ T cells (both $T_{CM}$ and $T_{EM}$) were drastically increased in the DLN and spleen of COF-919 + 660 + 808 nm treated mouse, indicating that COF-919-mediated phototherapy is favorable for promoting T-cell clone expansion (Fig. 7j, k, n, o). This was further evidenced by the immunohistochemical staining, where both CD8 and granzyme B, which play a critical role in cytotoxic T-cell-mediated antitumor immunity[58–60], were overexpressed in tumor tissues treated with COF-919 + 660 + 808 nm (Supplementary Fig. 19). Moreover, the proportions of immunosuppressive cells, including MDSCs and Treg cells, in DLN and spleen of COF-919 + 660 + 808 nm treated mouse were also sharply decreased (Fig. 7i, l, m, p), revealing that the COF-919-mediated phototherapy can relieve the immunosuppressive status and effectively reshape the tumor immune microenvironment for effective immunotherapy. Collectively, these results revealed that the COF-919-mediated phototherapy efficiently boosted the T-cell-mediated immune response.

**Synergistic effect of COF-919 and αPD-1**
Encouraged by the above in vivo therapeutic results, we suspected that COF-919-mediated phototherapy may be capable of improving the immune response rate of checkpoint blockade therapy. Therefore, a

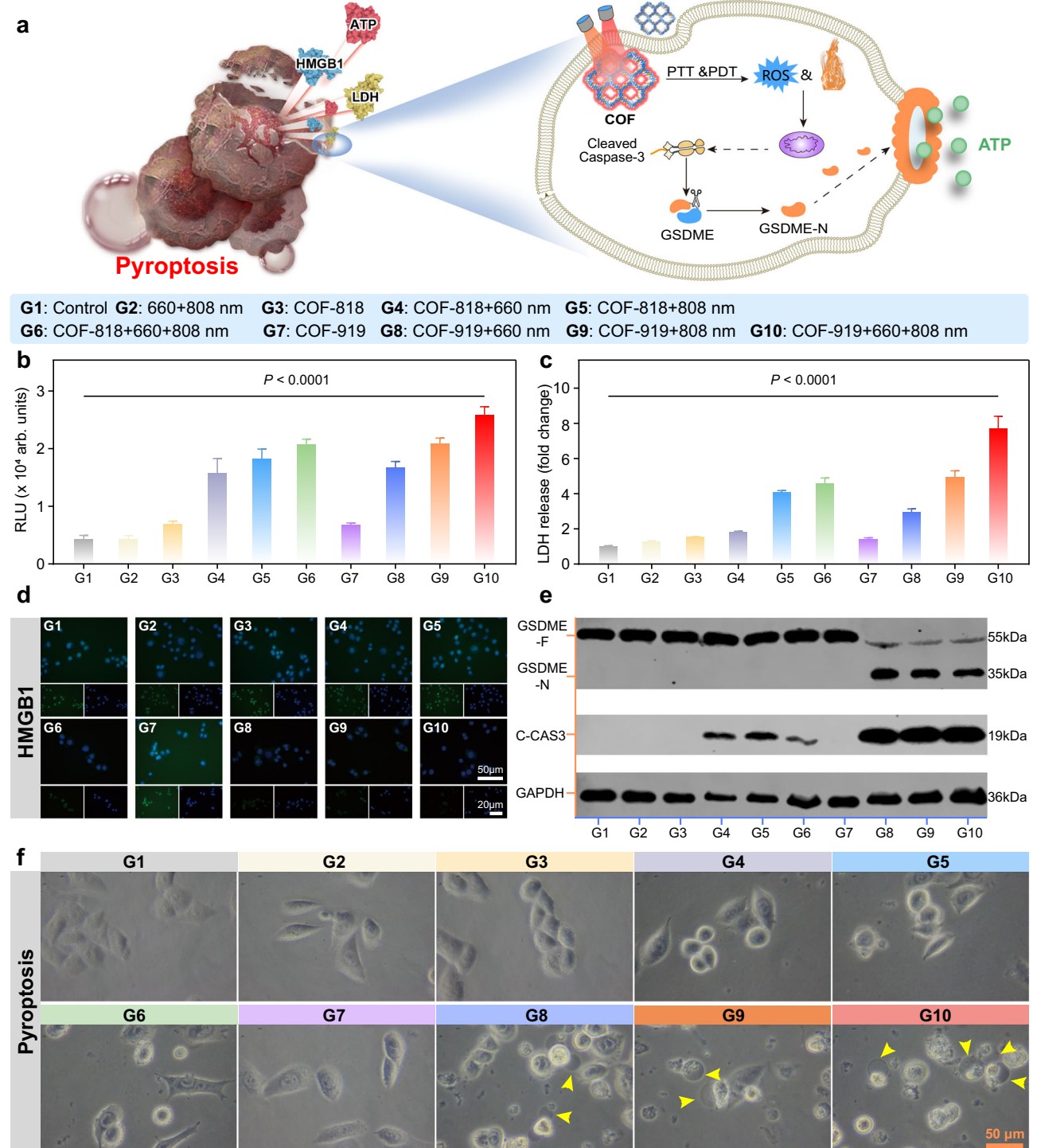

**Fig. 5 | Characterization of the pyroptosis-eliciting capability of COF-818 and COF-919. a** illustration of the mechanism of GSDME-dependent pyroptosis induced by COF-818 and COF-919. Quantitative evaluation of the representative pyroptosis factors, ATP (**b**) and LDH (**c**) in 4T1 cells after various treatments; data are presented as mean ± SEM (*n* = 3 independent samples). **d** Representative images of HMGB1 immunofluorescence staining in 4T1 cells after treatment with different samples. **e** Evaluate the expression levels of cleaved Caspase-3 and GSDME-N in 4T1 cells through western blot analysis; Western blot was done thrice independently with similar results. **f** Representative images of 4T1 cell morphology change after various treatments; the photographs of cell morphology were representative of those generated from three independent samples, each group with similar results; scale bar = 50 μm. Statistical significance was calculated via one-way ANOVA with Tukey's multiple comparisons test. Source data are provided as a Source Data file.

4T1 tumor-bearing dual-flank model was set up with PD-1 served as the immune checkpoint inhibitor (Fig. 8a). As seen in Figs. 8b, c, although the primary tumor was inhibited by COF-919 or COF-919 + αPD-1 treatment, the distant tumor could be suppressed only after treatment with COF-919 + αPD-1, demonstrating the great power of the

synergistic effect of COF-919 plus αPD-1. Furthermore, the tumor volume of the mice was examined using in vivo bioluminescence imaging every week. The data showed that COF-919 + αPD-1 combination therapy drastically prolonged the survival time of the 4T1 mice, where all of the COF-919 + αPD-1 treated mice survived euthanasia

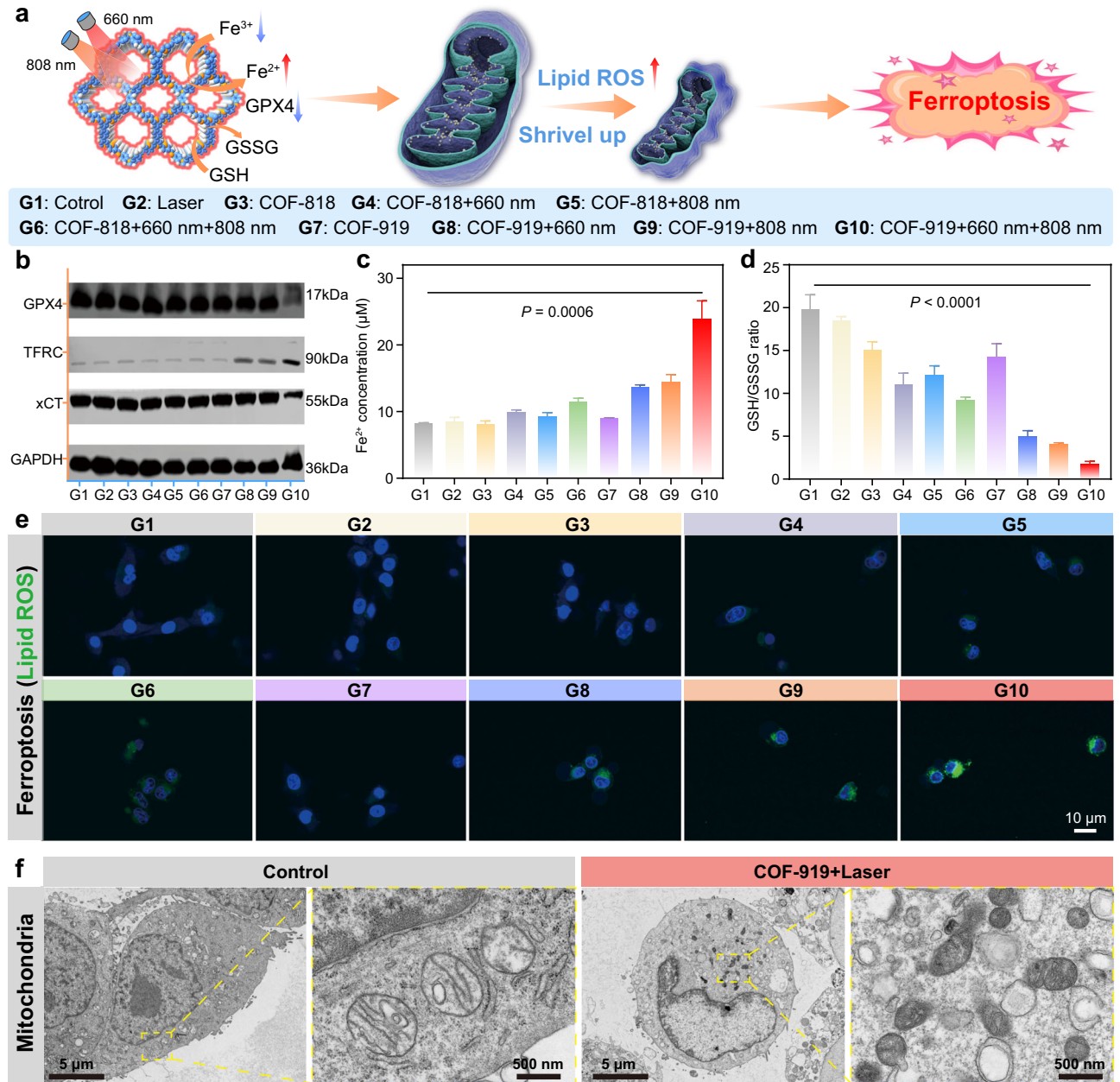

**Fig. 6 | Characterization of the ferroptosis eliciting capability of COF-818 and COF-919. a** Illustration the mechanism of GPX4-related ferroptosis induced by COF-818 and COF-919. **b** Western blot results of representative pyroptosis factors, including GPX4, TFRC, and xCT in 4T1 cells treated under different conditions. Quantitative evaluation of the changes in cellular iron concentration (**c**) and GSH/GSSG ratio (**d**), treated under different conditions; data are presented as mean ± SEM (*n* = 3 independent samples). **e** Confocal fluorescence images of 4T1 cells, where the green fluorescence was attributed to BODIPY 581/591-C11 staining of lipid peroxidation; the photographs of immunofluorescence staining were representative of those generated from three independent samples with similar results. **f** Bio-TEM images of mitochondria in normal cells or that of COF-919 treatment; the photographs of TEM were representative of those generated from three independent samples with similar results. Statistical significance was calculated via one-way ANOVA with Tukey's multiple comparisons test. Source data are provided as a Source Data file.

(Fig. 8d, e). In contrast, the control group showed a 30-day median survival time. To further investigate the potential of inhibiting tumor recurrence, a rechallenge 4T1 mouse model was established (Fig. 8f). As shown in Fig. 8g, h, the naïve mice suffered grave tumor recurrence, whereas the mice treated with COF-919 + αPD-1 showed no noticeable tumor recurrence. This indicated that this synergistic strategy of COF-919 + αPD-1 was favorable for restraining tumor recurrence.

## Discussion

In summary, a non-metallic AIE COF-based pyroptosis and ferroptosis dual-inducer was constructed by integrating planar and twisted motifs into a COF skeleton. Unlike traditional COFs constructed with solely planar or twisted motifs, this planar plus twisted AIE COF displayed better NIR light absorption, a lower band energy, and a longer lifetime which facilitate ROS generation and photothermal conversion, thus triggering a GSDME-dependent pyroptosis. Furthermore, owing to its exceptional ROS production capacity, COF-919 also upregulates intracellular lipid peroxidation, leading to glutathione depletion, low expression of glutathione peroxidase 4, and induction of a GPX4-related ferroptosis. Notably, COF-919-induced pyroptosis and ferroptosis effectively reshape the TME by promoting T-cell infiltration and alleviating immunosuppression, thereby

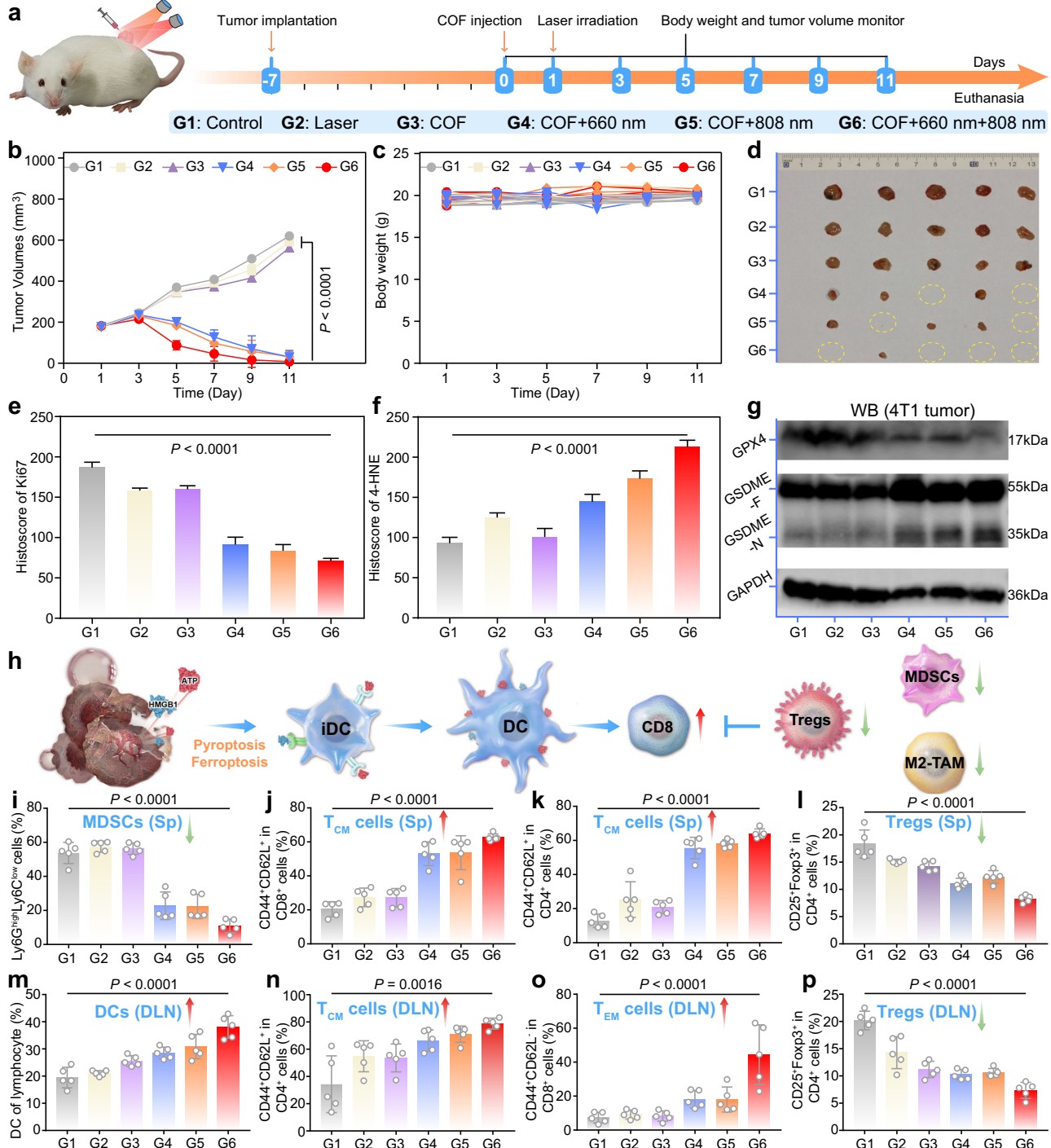

**Fig. 7 | Investigation of the in vivo therapeutic effect of COF-919-mediated phototherapy. a** Treatment schedule of COF-919-mediated phototherapy. Tumor volumes (**b**), body weights (**c**), and tumor images (**d**) of 4T1 tumor-bearing mice treated under different conditions; data are presented as mean ± SEM ($n = 5$ independent samples). Quantitative evaluation of Ki67(**e**) and 4-HNE (**f**); data are presented as mean ± SEM ($n = 5$ independent samples). **g** Western blot analysis was performed to determine the GPX4 and GSDME-N expression levels in tumor tissues; Western blot was done thrice independently with similar results. **h** Illustration of COF-919-mediated phototherapy to promote DC maturation. Quantitative evaluation of DCs in the DLN (**m**); $T_{CM}$ (**j**), and $T_{EM}$ (**k**) cells in the spleen; $T_{CM}$ (**n**) and $T_{EM}$ (**o**) cells in DLN; Tregs cells in the spleen (**l**) and DLN (**p**); MDSCs (**i**) cells in the spleen; data are presented as mean ± SEM ($n = 5$ independent samples). Statistical significance was calculated via two-way ANOVA (**b**) and one-way ANOVA with Tukey's multiple comparisons test (**e**, **f**, **i**–**p**). Source data are provided as a Source Data file.

boosting T-cell-mediated immune responses that are conducive to inhibiting tumor metastasis and recurrence. This work not only broadened the biomedical applications of AIE COFs but also provided an additional strategy for pyroptosis and ferroptosis dual-inducer design, which will benefit the further advancement in the field of cancer immunotherapy.

## Methods

### Synthesis of COF-818 and COF-919

A Pyrex tube was charged with M-TPh (77 mg, 0.12 mmol) or M-Tpy (78.5 mg, 0.12 mmol) and M-TPA (57 mg, 0.12 mmol). Next, 1.5 mL of o-dichlorobenzene and n-butanol were added along with 0.3 mL aqueous acetic acid (6 M). Then, the tube was swiftly sealed by flame and

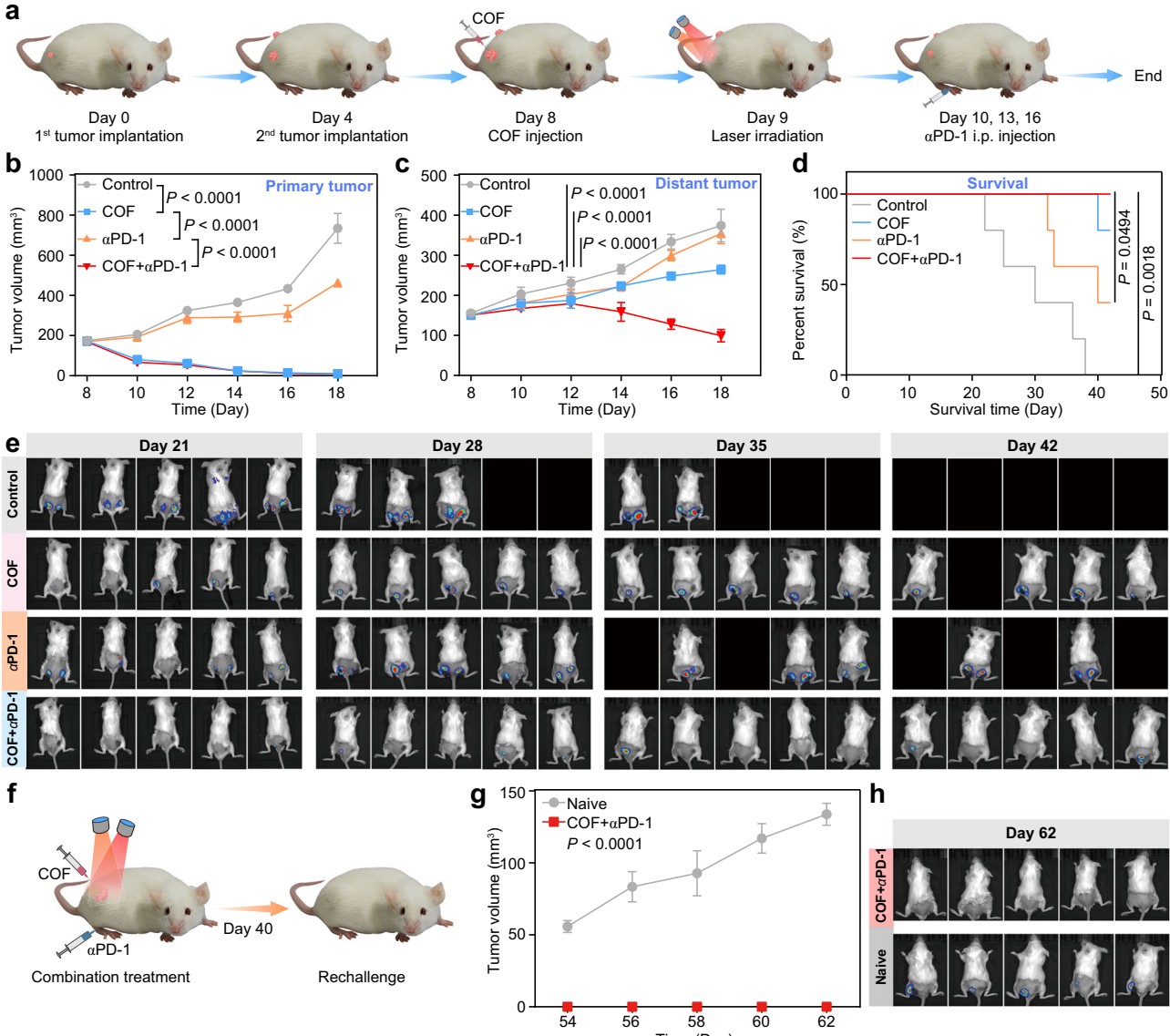

**Fig. 8 | The synergistic effect of COF-919 + αPD-1. a** Schedule of COF-919 + αPD-1 combination therapy. The growth curves of primary (**b**), distant tumor (**c**), and percent survival (**d**) of 4T1 tumor-bearing mice with different treatments; data are presented as mean ± SEM ($n = 5$ independent samples). **e** Bioluminescence images of 4T1 tumor-bearing mice at different times; data are presented as mean ± SEM ($n = 5$ independent samples). **f** Establishment of the rechallenge model. Tumor volume curves (**g**) and bioluminescence images (**h**) of naive and COF-919 + αPD-1 treatment groups at different times; data are presented as mean ± SEM ($n = 5$ independent samples). Statistical significance was calculated via two-way ANOVA with Tukey's multiple comparisons test (**b**, **c**, **g**) and log-rank (Mantel–Cox) test (**d**). Source data are provided as a Source Data file.

subjected to a 120 °C oven for 3 days. The ensuing products were filtered and activated with dioxane and acetone. These AIE COFs were dried under a dynamic vacuum at 120 °C for 24 h to attain complete activation.

## Cell and animals

The murine breast cancer cell line 4T1 (Cat CRL-2539) was purchased from the American Type Culture Collection (ATCC, Manassas, VA). For culturing 4T1 cells, a medium containing RPMI-1640 (Gibco), 0.5 mg mL$^{-1}$ of penicillin–streptomycin (Gibco), and 10% Certified Fetal Bovine Serum, FBS (VivaCell, Shanghai, China) was employed. The culture conditions of 4T1-Luc cells were maintained similarly to that of 4T1 cells, albeit with the medium supplemented with 1 μg mL$^{-1}$ of puromycin (Invitrogen). The female BALB/c mice were provided by the Experimental Animal Central of Wuhan University. We used female mice because mammary cancers occur primarily in females. The in vivo

experiments were conducted in accordance with the Institutional Animal Care and Use Committee (IACUC) guidelines of Wuhan University (approval number WP20220030). All animal experimental procedures were performed in accordance with the Regulations for the Administration of Affairs Concerning Experimental Animals approved by the State Council of the People's Republic of China. All mice were housed under specific pathogen-free (SPF) conditions (temperature ~22 °C, humidity ~50%) with a 12/12 h dark/light cycle. 4T1 ($7 \times 10^5$) cells resuspended in 100 μL of RPMI medium were subcutaneously injected into each BALB/c mouse. Tumor volume was calculated as follows: width$^2$ × length × 0.5. All mice in this study were euthanized under $CO_2$ anesthesia if the volume of the primary tumor reached a maximum allowable volume of 1500 mm$^3$ or if the tumor burden compromised the animal welfare. The maximal tumor size in this study was not exceeded. All animal experiments were approved by the Wuhan University Animal Experiment Ethics Committee.

## Cytotoxicity assay

4T1 cells ($3 \times 10^3$) were incubated overnight in RPMI-1640 medium, following which the culture medium was substituted with varying concentrations (0, 10, 30, 50, 70, 90 µg mL$^{-1}$) of COF materials and irradiated under 660 nm (0.5 W cm$^{-2}$, 5 min) and/or 808 nm (1.5 W cm$^{-2}$, 5 min) lasers. After 24 h of incubation, the viability of 4T1 cells was assessed using MTT.

## Pyroptosis assay

4T1 cells ($2 \times 10^4$) were incubated with COF samples (100 µg mL$^{-1}$) for 24 h. The resulting samples were subjected to different treatments and irradiated under 660 nm (0.5 W cm$^{-2}$, 5 min) and/or 808 nm (1.5 W cm$^{-2}$, 5 min) lasers. The cell supernatants were collected after 6 h, and the extracellular release of ATP was measured using an ATP assay kit (S0026, Beyotime Biotechnology). The experimental LDH release was determined using an LDH release kit (C0016, Beyotime Biotechnology).

## Intracellular detection of HMGB1

Different concentrations of COF samples were added to 4T1 cells and incubated overnight. Following various treatments, the 4T1 cells ($2 \times 10^3$) were fixed with 4% paraformaldehyde, infiltrated with 0.5% Triton X-100 (BS084, Biosharp), blocked with 3% BSA (Biosharp BS114) in 0.1% triton/PBS for 1 hour, and incubated with anti-HMGB1 antibodies (1:400, ab18256, Abcam) at room temperature and Goat anti-Rabbit IgG DyLight 488 (1:200, A23220, Abbkine). The cells were then counterstained with DAPI (Beyotime Biotechnology, P0131), The slides were examined with an FV1000 confocal microscope (Olympus) using FV10-ASW software v. 4.0 (Olympus). The fluorescence signals were analyzed using Image-Pro Plus software v. 6.0 (Media Cybernetics).

## Photothermal properties

Totally, 1.0 mL of COF samples with different concentrations were added and irradiated by an 808 nm NIR laser (Beijing Laserwave Optoelectronics Technology Co., Ltd.). An infrared camera (FOTRIC 225s, China) was utilized to monitor the temperature change in real time.

## Ferroptosis assay

4T1 cells ($5 \times 10^4$) and COF samples (100 µg mL$^{-1}$) were incubated for 24 h, then irradiated with 660 nm (0.5 W cm$^{-2}$, 5 min) and/or 808 nm (1.5 W cm$^{-2}$, 5 min) lasers. Then, the cells were used for GSH/GSSG assay. The ratio of GSH/GSSG in cells was measured using the GSH/GSSG assay kit (Beyotime Biotechnology, S0053) as per the manufacturer's instructions. Total GSH and oxidized GSH were quantified using a luminescence reader. Totally, $5 \times 10^4$ 4T1 cells were collected and washed twice with PBS from each group. After treatment, according to the manufacturer's instructions (E1042, APPLYGEN, Beijing), the supernatant was collected, and the absorbance value of 550 nm was detected by the microplate reader and zeroized with a blank hole.

## Measurement of lipid ROS

C11-BODIPY (581/591) dye (D3861, Invitrogen) was employed as a probe to assay the lipid ROS level, which is readily oxidized by lipid ROS, leading to a gradual increase in fluorescent intensity. Cells from each experimental group were washed and incubated in 1 mL of PBS containing 0.5 µM C11-BODIPY at 37 °C for 60 min. The fluorescence signal change was evaluated using confocal microscopy.

## Western blot

The total proteins were lysed using a lysate buffer (Beyotime, Shanghai, China) for 30 minutes and washed with PBS. Polyacrylamide gel electrophoresis was employed to separate the protein, which was then loaded onto a polyvinylidene fluoride membrane (Millipore). Non-specific antigens were blocked with 5% skim milk for 2 h, followed by immunoblotting with different antibodies, including GSDME (1:1000, ab215191, Abcam), cleaved Caspase-3 (1:1000, #9664, CST), Glutathione Peroxidase 4 (1:1000, ab125066, Abcam), Transferrin Receptor (1:1000, ab214039, Abcam), xCT (1:1000, T57046, Abmart), GAPDH (1:5000, 60004-1-Ig, Proteintech).

## Immunohistochemistry

Following euthanasia, tumor tissues from each group were collected and fixed with 4% paraformaldehyde. The tissues were then embedded in paraffin and cut into 4 µm sections, which were subsequently deparaffinized and rehydrated. Antigen retrieval was performed at high temperatures in a pressure cooker using citrate buffer (pH 6.0). To prevent the non-specific binding of antibodies, goat serum was employed. Primary antibodies, including Ki67 (1:400, ab15580, Abcam), CD8 (1:400, #98941, CST), Granzyme B (1:200, #44153, CST), cleaved Caspase-3 (1:1000, #9664, CST), and 4-HNE (1:400, ab46545, Abcam), were used for the immunohistochemistry staining. The panoramic DESK digital pathology scanner (3D HISTECH) was used to scan all sections.

## Antitumor efficacy of COF materials

To establish a 4T1 mouse model, the right flank of the BALB/c mouse was injected with $1 \times 10^6$ 4T1 cells. The model was then randomly assigned to six groups, including PBS, 660 nm (0.5 W cm$^{-2}$, 5 min) plus 808 nm (1.5 W cm$^{-2}$, 5 min), COF-919, COF-919 + 660 nm (0.5 W cm$^{-2}$, 5 min), COF-919 + 808 nm (1.5 W cm$^{-2}$, 5 min), and COF-919 + 660 nm (0.5 W cm$^{-2}$, 5 min) + 808 nm (1.5 W cm$^{-2}$, 5 min). The injected concentration of COF was 100 µg mL$^{-1}$ (100 µL) per mouse.

## Abscopal effect

PD-1 was selected as the immune checkpoint inhibitor. To establish a 4T1 tumor-bearing dual-flank model, $1 \times 10^6$ 4T1-Luc cells were injected, following which the model was divided into four groups, including control, COF-919 + Laser, αPD-1, and COF-919 + Laser + αPD-1 groups. On days 2, 5, and 8, αPD-1 antibody was injected intraperitoneally (5 mg kg$^{-1}$). Tumor volume was measured every two days, and the mice were euthanized when ethical endpoints were reached.

## Rechallenge study

To investigate the immune memory of COF-919+Laser+αPD-1, a 4T1 tumor-bearing mice model was set up. COF-919 + Laser + αPD-1 treatments were performed when the tumors reached a volume of 100 mm$^3$. After 40 days, both the COF-919 + Laser + αPD-1 treated and naive groups were re-inoculated with $3 \times 10^5$ 4T1-Luc cells.

## Flow cytometry analysis

Single-cell suspensions of splenic and inguinal nodes were prepared by perfusing mice with PBS, followed by staining with various antibodies and analyzing using flow cytometry on FACS caliber (Beckman). The eFluor 506 dye (eBioscience) was utilized to exclude dead cells. The stained cells were run on a flow cytometer (Beckman) using the CytExpert software v. 2.3 (Beckman) and then analyzed by FlowJo software (v. 10, TreeStar).

## Statistical analysis

Data are represented as the mean ± standard error of the mean (s.e.m.) or standard deviation (s.d.) as indicated in the figure legends. One-way ANOVA with Tukey's multiple comparisons was used for multiple comparisons when more than two groups were compared, and two-tailed Student's $t$-test was used for two-group comparisons. The survival benefit was determined using a log-rank (Mantel−Cox) test. All statistical tests were performed using GraphPad Prism software v. 8.0 (GraphPad Software) and Excel 2016 software (Microsoft). In all types of statistical analysis, values of $P < 0.05$ were considered significant.

## Reporting summary

Further information on research design is available in the Nature Portfolio Reporting Summary linked to this article.

## Data availability

The data generated in this study are available within the Article, Supplementary Information, or Source Data file. Source data are provided in this paper. The full image dataset is available from the corresponding author upon request. Source data are provided in this paper.

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

## Acknowledgements
This work is supported by the National Natural Science Foundation of China (82273202, 82002879, 82072996, 21788102, 22025106, 91545205, 91622103, and 21971199), the Research Grants Council of Hong Kong (16305320, 16306620, N-HKUST609/19 and C6014-20W), the Innovation and Technology Commission (ITCCNERC14SC01), National Key Research and Development Program of China (2018YFA0704000 and 2022YFC2504200), National Key Basic Research Program of China (2014CB239203), Innovation Team of Wuhan University (2042017kf0232), China Postdoctoral Science Foundation (2021M692475, 2021T140524, and XJ2021037), and Interdisciplinary innovative foundation of Wuhan University (XNJC202303). The authors would like to thank Jilin Chinese Academy of Sciences-Yanshen Technology Co., Ltd. for the supply of ligands, Yuan Zhou from Shiyanjia Lab (www.shiyanjia.com) for their DFT calculations, and Dr. Xue Zhou from the Core Research Facilities of CCMS (WHU) for her assistance with Solid-State NMR analysis.

## Author contributions
Z.-J.S., B.Z.T., H.D., and J.W.Y.L. conceived and designed the experiments. L.Z., A.S., Q.-C.Y., S.W., and S.-C.W. performed the experiments. L.Z., R.T.K.K., and J.S. analyzed the data. L.Z., A.S., S.J.L., and Z.-J.S. wrote the paper. All authors have given approval to the final version of the paper.

## Competing interests
The authors declare no competing interests.
