## [Peer Review File · Nature Communications]

Reviewers' Comments:

Reviewer #1:

Remarks to the Author:

The authors developed COF-919 by incorporating planar and twisted motifs into a COF skeleton which could induce pyroptosis and ferroptosis to elicit strong inflammatory response for cancer immunotherapy. The manuscript containing interesting results merits publication. However, the manuscript needs further improvement to meet the publication requirements of this journal. Some questions and suggestions are as follows:

1. The structure of COF 919 appears insoluble in water. How much organic solvent is in the administration solution? The same proportion of organic solvent should be considered as control group.
2. Were the two lasers irradiated simultaneously or in a particular order in the cell and animal experiments? Besides, the 660 nm laser also appears to have photothermal capabilities, so why use two lasers? A change in laser power may have a similar effect.
3. As the decrease in GPX4 and xCT (Fig. 6b) treated with some groups is not clearly evident, please supplement the quantitative data.
4. Did mice receive tumor injections? If so, why not irradiate after the drug administration but one day after?
5. There are many typos, format issues, and detail errors in the manuscript, particularly in the caption. Authors should carefully check again.

Reviewer #2:

Remarks to the Author:

Aggregation-induced emission (AIE)-based compounds are the most extensively studied molecules. Over the past decade, great efforts have been invested in exploring the covalent organic framework (COF) field. COFs are a unique class of porous materials with extended molecular frameworks. Their structures can be pre-designed via the topological design of building blocks, and the organization of extended structures is driven by either reversible or irreversible chemical reactions to assume crystallinity and structural integrity. The applications of COFs have been widespread in different fields majorly including semiconductors, luminescence and sensors, energy storage, adsorption and separation, catalysis and drug delivery. Interestingly this class of materials have not been much explored in the field of cancer therapy. In the present manuscript, authors have designed and developed a non-metallic COF-based pyroptosis and ferroptosis dual-inducer by introducing AIEgens into COF skeletons, to achieve a strong inflammatory response and boost antitumor immunity. The synthesised AIE COF-919 is found to be superior to the traditional COF-909 and COF-818. It displayed better near-infrared light absorption, lower band energy and a longer lifetime to contribute a good phototherapeutic effect, which is favourable for inducing pyroptosis. Traditional COF-based photosensitizers (PSs) suffer from strong undesirable aggregation-caused quenching (ACQ) effects which limit the generation of reactive oxygen species (ROS) ultimately failing to trigger pyroptosis or ferroptosis. But the reported AIE COF-919 is not suffered from ACQ effects and showed outstanding ROS production capability, intracellular lipid peroxidation (LPO) is also upregulated, which leads to glutathione depletion and induction of GPX4-related ferroptosis.

- It is essential to revise the language component of the manuscript. For instance, line 212, "in 4T1 cells was observed in 4T1 cells treated". Line 206, "sufficient ROS in lipids will result in LPO".
- Line 265, "Pyroptosis and ferroptosis combination therapy". This is not giving the intended meaning. I assume it should sound like "Ferroptosis/pyroptosis dual-inductive combinational anti-cancer therapy". Many such errors can be noticed throughout the text.
- The results need to be discussed in detail.

Reviewer #3:

Remarks to the Author:

This manuscript describes the development of a COF-919 from planar and twisted AIEgens motifs

which induced pyroptosis and ferroptosis and triggered strong anti-inflammatory response for efficient antitumor efficiency. This manuscript needs major revisions.

1. The rationale behind using AIEgens as monomeric units to form the COF-919 is not clear as the authors never used the AIEgen activity of for bioimaging in vitro or in vivo.

2. For the monomeric AIEgens, the authors seriously lacked the characterization:

(a) UV-Vis spectra of the monomers should be provided to confirm the absorbance at NIR region (Fig. 2).

(b) What is the Stokes shift for the monomers?

(c) Wave length of the emission needs to be reported.

(d) What is the underlying mechanism of the AIE property of the monomers?

3. The AIE property of the COF-919 and COF-818 is not characterized properly:

(a) why there is a red shift for COF-919 vs COF-818 (Fig. 3b)?

(b) The UV-Vis spectra are very broad and the absorbance is higher in sub 500 nm and decreased in 800nm which does not justify the NIR absorbance.

(c) How much is the Stokes shift for the COF-919 and COF-818 as that will be important for the bioimaging.

(d) No bioimaging data is provided to justify the incorporation of AIEgens in the COFs.

(e) What is the quantum yield of the COF-919 and COF-818?

(f) What is the mechanism of the AIE property in the COF-919 and COF-818?

(g) The AIE property of COF-919 and COF-818 is missing. Authors should show the binary solvent dependent change in fluorescence by titration curve.

(h) The authors should show the images of the COFs in different binary solvents under UV light to show the AIE property also show the AIE property in the solid state.

4. (a) What is the IC50 value of COF-818 and COF-919 in the cells in presence of light and dark?

(b) It is not mentioned in the manuscript what is the power and time of the laser used for the cell viability assay?

(c) What is the concentration of COFs used for the cell viability and the ROS generation assays?

(d) Why nanoscale COF samples were generated by ultrasound treatment (Fig. 4j)?

(e) The authors should perform MTT assay to evaluate the cell viability in presence of laser and dark.

(f) Control cells should be treated with only lasers to show that cells are not killed by the laser itself.

(g) What is the concentration of the COFs used for pyroptosis and ferroptosis assays?

(h) What was the power and time of the laser irradiation used for the pyroptosis and ferroptosis assays?

(i) For the in vivo studies, what is the dose of the COFs used per mice and what was the power of the laser used for how long?

Point-by-Point Response to Reviewers' Comments

Reviewers Comments:

=====
Reviewer #1:

Comments:

The authors developed COF-919 by incorporating planar and twisted motifs into a COF skeleton which could induce pyroptosis and ferroptosis to elicit strong inflammatory response for cancer immunotherapy. The manuscript containing interesting results merits publication. However, the manuscript needs further improvement to meet the publication requirements of this journal.

Response:

We would like to thank this reviewer for the strong support on the main findings of this work, and constructive suggestions that helped us to further the quality. By responding to the reviewer's comments in detail and revising the manuscript accordingly, we believe our manuscript has been significantly strengthened. All revisions are highlighted in **BLUE** color in the revised manuscript and Supplementary Information.

Comment 1:

1. "The structure of COF-919 appears insoluble in water. How much organic solvent is in the administration solution? The same proportion of organic solvent should be considered as control group"

Response 1:

We thank this reviewer for the constructive suggestion. Actually, porous materials such as MOFs and COFs are commonly known to be insoluble in any solvents. Therefore, in order to achieve a well-dispersed COF nanoparticle system, sonication was employed using an ultrasonic cell shredder. A detailed methodology utilized for COF nanoparticle dispersion can be found in the Supplementary Information, page 2.

Figure R1. Illustration of the strategy of disperse microsize COF sample to nanosize COF sample via sonication.

Comment 2:

2. “Were the two lasers irradiated simultaneously or in a particular order in the cell and animal experiments? Besides, the 660 nm laser also appears to have photothermal capabilities, so why use two lasers? A change in laser power may have a similar effect.”

Response 2:

We thank this reviewer for the careful observations. In this work, the COF samples were subjected to two rounds of irradiation, first under a 660 nm laser at a power density of 0.5 W cm⁻², and subsequently under an 808 nm laser at a power density of 1.5 W cm⁻². Owing to the restricted power density of the 660 nm laser, which is typically less than 1 W cm⁻², the photothermal effect exhibited by COF-818 and COF-919 was found to be inadequate (**Figure R2**). To achieve desirable photothermal capabilities, an 808 nm laser with a power density of 1.5 W cm⁻² was employed.

Figure R2. Temperature changes in the control, COF-818 and COF-919 samples under 808 or 660 nm laser irradiation.

Comment 3:

3. “As the decrease in GPX4 and xCT (Fig. 6b) treated with some groups is not clearly evident, please supplement the quantitative data.”

Response 3:

We sincerely thank this reviewer for the careful observations. As the reviewer suggested, we repeat this experiment carefully, and have included the quantitative data to support our findings, with details as following:

Figure R3. Western blot results of representative ferroptosis factors, including GPX4, TFRC and xCT in 4T1 cells treated under different conditions.

Revision made:

We have added the data of Fig. R3 as “Supplementary Fig. S16” in the Supplementary Information and highlighted in **BLUE** on page 19.

Comment 4:

4. “*Did mice receive tumor injections? If so, why not irradiate after the drug administration but one day after?*”

Response 4:

We sincerely thank the reviewer for such careful reading. Indeed, the mice were subjected to tumor injection, and the rationale behind the one-day delay in irradiation primarily pertains to the diffusion and penetration of COF samples throughout the entire tumor. The temporal requirement for the diffusion process is supported by the confocal fluorescence image results (**Figure R4**), similar experimental design has also been employed in other published works (*Adv. Funct. Mater.* **2022**, 2201542; *Adv. Mater.* **2022**, 2108174).

Figure R4. IVIS imaging of 4T1 tumor-bearing mouse after injected with COF sample for different times.

Revision made:

We have added the data of Fig. R4 as “Supplementary Fig. S12” in the Supplementary Information and highlighted in **BLUE** on page 17.

Comment 5:

5. “*There are many typos, format issues, and detail errors in the manuscript, particularly in the caption. Authors should carefully check again.*”

Response 5:

We sincerely thank the reviewer for such careful reading. As the reviewer suggested, we revised the manuscript carefully and highlight in **BLUE** in the revised manuscript.

Reviewer #2:

Comments:

Aggregation-induced emission (AIE)-based compounds are the most extensively studied molecules. Over the past decade, great efforts have been invested in exploring the covalent organic framework (COF) field. COFs are a unique class of porous materials with extended molecular frameworks. Their structures can be predesigned via the topological design of building blocks, and the organization of extended structures is driven by either reversible or irreversible chemical reactions to assume crystallinity and structural integrity. The applications of COFs have been widespread in different fields majorly including semiconductors, luminescence and sensors, energy storage, adsorption and separation, catalysis and drug delivery. Interestingly this class of materials have not been much explored in the field of cancer therapy. In the present manuscript, authors have designed and developed a non-metallic COF-based pyroptosis and ferroptosis dual-inducer by introducing AIEgens into COF skeletons, to achieve a strong inflammatory response and boost antitumor immunity. The synthesised AIE COF-919 is found to be superior to the traditional COF-909 and COF-818. It displayed better near-infrared light absorption, lower band energy and a longer lifetime to contribute a good phototherapeutic effect, which is favourable for inducing pyroptosis. Traditional COF-based photosensitizers (PSs) suffer from strong undesirable aggregation-caused quenching (ACQ) effects which limit the generation of reactive oxygen species (ROS) ultimately failing to trigger pyroptosis or ferroptosis. But the reported AIE COF-919 is not suffered from ACQ effects and showed outstanding ROS production capability, intracellular lipid peroxidation (LPO) is also upregulated, which leads to glutathione depletion and induction of GPX4-related ferroptosis.

Response:

We are very grateful to the reviewer for their careful reading and pointing out the novelty of our work. We also highly appreciate the reviewer's suggestion for strengthening our work. By responding to the reviewer's comments in detail and revising the manuscript accordingly, we believe that our manuscript has been significantly strengthened. All revisions are highlighted in **BLUE** color in the revised manuscript and Supplementary Information.

Comment 1:

1. *“It is essential to revise the language component of the manuscript. For instance, line 212, “in 4T1 cells was observed in 4T1 cells treated”. Line 206, “sufficient ROS in lipids will result in LPO”*

Response 1:

We thank this reviewer for the careful observations. As the review suggested, we revised the manuscript carefully and highlight in **BLUE** in the revised manuscript.

Comment 2:

2. “Line 265, “Pyroptosis and ferroptosis combination therapy”. This is not giving the intended meaning. I assume it should sound like “Ferroptosis/pyroptosis dual-inductive combinational anti-cancer therapy”. Many such errors can be noticed throughout the text.”

Response 2:

We thank this reviewer for the careful reading. The incorrect statement has been revised and highlighted in BLUE on page 13, with details as following:

“As immunogenic programmed cell death modes, pyroptosis and ferroptosis has demonstrated its potential to induce an acute inflammatory response to produce antitumor immune activity, and boost the cancer immunotherapy (Fig. 7h).”

Comment 3:

3. “The results need to be discussed in detail.”

Response 3:

Thanks for the reviewer’s insightful comment. As the review suggested, we have incorporated a more comprehensive and detailed discussion in the revised manuscript (page 16 to 17) and highlight in BLUE, with details as following:

Unlike traditional COFs constructed with solely planar or twisted motifs, this planar plus twisted AIE COF displayed better NIR light absorption, a lower band energy and a longer lifetime which facilitate ROS generation and photothermal conversion, thus triggering a GSDME-dependent pyroptosis. Furthermore, owing to its exceptional ROS production capacity, COF-919 also upregulates intracellular lipid peroxidation, leading to glutathione depletion, low expression of glutathione peroxidase 4, and induction of a GPX4-related ferroptosis. Notably, COF-919-induced pyroptosis and ferroptosis effectively reshape the TME by promoting T cell infiltration and alleviating immunosuppression, thereby boosting T-cell-mediated immune responses that are conducive to inhibiting tumor metastasis and recurrence.

Reviewer #3:

Comments:

This manuscript describes the development of a COF-919 from planar and twisted AIEgens motifs which induced pyroptosis and ferroptosis and triggered strong anti-inflammatory response for efficient antitumor efficiency. This manuscript needs major revisions.

Response:

We would like to thank the reviewer for the positive and constructive comments. We also highly appreciate the reviewer's suggestion for strengthening our work. We have performed more experiments to address the reviewer's concerns. By responding to the reviewer's comments in detail and revising the manuscript accordingly, we believe this manuscript has been strengthened. All revisions are highlighted in BLUE color in the revised manuscript and Supplementary Information.

Comment 1:

1. "The rationale behind using AIEgens as monomeric units to form the COF-919 is not clear as the authors never used the AIEgen activity of for bioimaging *in vitro* or *in vivo*."

Response 1:

We thank the reviewer for this constructive suggestion. We added the *in vitro* and *in vivo* bio-images of COF-919 in the revised Supplementary Information and highlighted in BLUE, with details as following:

Figure R5. The *in vitro* and *in vivo* bioimaging of COF-919.

Revision made:

We have added the data of Fig. R5 as "Supplementary Fig. S12" in the Supplementary Information and highlighted in BLUE on page 17.

Comment 2a:

2a. “For the monomeric AIEgens, the authors seriously lacked the characterization: (a) UV-Vis spectra of the monomers should be provided to confirm the absorbance at NIR region (Fig. 2)”

Response 2a:

Thanks for the reviewer’s insightful comment. As the reviewer suggested, we conducted UV-Vis and fluorescence spectral analysis on the monomers and subsequently calculated their respective Stokes shifts, while also labeling their emission wavelengths (**Figure R6**), with details as following:

Figure R6. The UV-Vis spectra, fluorescence spectra, and Stokes shift of monomers, including M-TPy, M-TPh, and M-TPA.

Revision made:

We have added the data of Fig. R6 as “Supplementary Fig. S10” in the Supplementary Information and highlighted in **BLUE** on page 17.

Comment 2b:

2b. “(b) What is the Stokes shift for the monomers?”

Response 2b:

The Stokes shift of monomers, including M-TPy, M-TPh, and M-TPA, are determined to be 60, 70 and 120 nm, respectively (**Figure R6**).

Comment 2c:

2c. “(c) Wavelength of the emission needs to be reported.”

Response 2c:

The emission wavelength of the monomers, including M-TPy, M-TPh, and M-TPA, are found to be 367, 366 and 440 nm, respectively (**Figure R6**).

Comment 2d:

2d. “(d) What is the underlying mechanism of the AIE property of the monomers?”

Response 2d:

Thanks for the reviewer’s insightful comment. Among these monomers, M-TPh and M-TPy were found to exhibit classical AIE characteristics, while M-TPA displayed an ACQ phenomenon. Our previous studies have suggested that restriction of intramolecular motion (RIM) may serve as the underlying mechanism responsible for the AIE property of M-TPh and M-TPy. Specifically, it is believed that the emission of these monomers is rejuvenated by the prevention of non-radiative energy dissipation upon aggregate formation. The ACQ effect observed in M-TPA, on the other hand, may be attributed to its twisted configuration which

results in a strong twisted intramolecular charge transfer (TICT) effect, leading to significant non-radiative decay and fluorescence quenching.

Comment 3a:

3a. “The AIE property of the COF-919 and COF-818 is not characterized properly: (a) why there is a red shift for COF-919 vs COF-818 (Fig. 3b)?”

Response 3a:

We thank this reviewer for the careful observations. The observed red shift in COF-919 compared to COF-818 (Fig. 3b) may be attributed to the larger π -conjugation of the linkers within the entire COF-919 structure. Specifically, while the M-TPh linker exhibits a twisted structure, the M-TPy monomer adopts a more planar configuration due to the presence of hydrogen bonds. Consequently, the conjugation degree of COF-919 is higher than that of COF-818, leading to a smaller band gap and red shift in the UV-Vis absorption spectra. This observation is consistent with the results obtained from DFT calculations, wherein COF-919 also exhibits a smaller band gap (**Figure 3h to 3i**).

Figure R7. Illustration of the different configuration between COF-818 and COF-919.

Comment 3b:

3b. “(b) The UV-Vis spectra are very broad and the absorbance is higher in sub 500 nm and decreased in 800 nm which does not justify the NIR absorbance.”

Response 3b:

We acknowledge that in this study, the absorbance of COFs is higher in the sub 500 nm region and decreases in the 800 nm region, indicating suboptimal NIR absorption performance that limits the PDT and PTT efficacy. The construction of COF materials with excellent NIR absorption performance remains a challenge due to the lack of suitable NIR-absorbing monomers and strict synthesis conditions. We appreciate the reviewer's suggestion, which has motivated us to explore the feasibility of constructing NIR-absorbing COFs to enhance their PDT and PTT performance in future investigations.

Comment 3c:

3c. “(c) How much is the Stokes shift for the COF-919 and COF-818 as that will be important for the bioimaging.”

Response 3c:

As shown in **Figure R8**, the Stokes shift for COF-919 and COF-818 are determined to be 140 and 160 nm, respectively.

Figure R8. The UV-Vis spectra, fluorescence spectra, and Stokes shift of COF-818 and COF-919.

Revision made:

We have added the data of Fig. R8 as “Supplementary Fig. S11” in the Supplementary Information and highlighted in **BLUE** on page 17.

Comment 3d:

3d. “(d) No bioimaging data is provided to justify the incorporation of AIEgens in the COFs.”

Response 3d:

We thank the reviewer for this constructive suggestion. We added the *in vitro* and *in vivo* bio-images pictures of COF-919 in the revised Supplementary Information and highlighted in **BLUE**, with details as following:

Figure R5. The *in vitro* and *in vivo* bioimaging of COF-919.

Revision made:

We have added the data of Fig. R5 as “Supplementary Fig. S12” in the Supplementary Information and highlighted in **BLUE** on page 18.

Comment 3e:

3e. “(e) What is the quantum yield of the COF-919 and COF-818?”

Response 3e:

We thank this reviewer for the valuable suggestion. COF-818 and COF-919 were constructed by co-condensation of an AIE monomer (M-TPh or M-TPy) and an ACQ monomer (M-TPA). As a result, the brightness of these COFs was found to be partially quenched by M-TPA, ultimately leading to a low quantum yield. Specifically, the quantum yield of COF-919 was measured to be 1%, while that of COF-818 was determined to be 1.5%.

Comment 3f:

3f. “(f) What is the mechanism of the AIE property in the COF-919 and COF-818?”

Response 3f:

Thanks for the reviewer’s insightful comment. As the reviewer suggested, we tested the PL spectra of COF-818 and COF-919 in THF/H₂O solutions with different water fractions (**Figure R9**). Due to porous materials such as MOFs and COFs are commonly insoluble in any solvents, It’s hard to evaluate the AIE property of COF-818 and COF-919 using this method.

Figure R9. Photoluminescence (PL) spectra of COF-818 (A) and COF-919 (C) in THF/H₂O solutions with different water fractions (f_w). (C) Plots of the relative emission intensity (I/I_0) of COF-818 and COF-919 versus increased water fraction.

Comment 3g:

3g. “(g) The AIE property of COF-919 and COF-818 is missing. Authors should show the binary solvent dependent change in fluorescence by titration curve.”

Response 3g:

We thank the reviewer for this constructive suggestion. As the reviewer suggested, The AIE properties of COF-818 and COF-919 were investigated in mixed solutions of tetrahydrofuran (THF) and water (H₂O) with different H₂O ratios (**Figure R9**). The results showed that, by gradual increasing the H₂O ratio, the emission intensity of COF-818 and COF-919 were not changed obviously. This phenomenon might be general for AIEgens-based porous materials such as MOFs and COFs, due to its insoluble in any solvents.

Comment 3h:

3h. “(h) The authors should show the images of the COFs in different binary solvents under UV light to show the AIE property also show the AIE property in the solid state.”

Response 3h:

We thank this reviewer for the constructive suggestion. The images of the COFs in different binary solvents under UV light were inserted in **Figure R9**.

Comment 4a:

4a. “(a) What is the IC_{50} value of COF-818 and COF-919 in the cells in presence of light and dark?”

Response 4a:

We thank this reviewer for the valuable suggestion. In accordance with the reviewer's suggestion, we conducted additional MTT experiments in 4T1 cells to determine the IC_{50} value of COF-818 and COF-919. The results indicated that in the absence of laser irradiation, even with a concentration of $90 \mu\text{g mL}^{-1}$, more than half of the cells still alive, which demonstrated that toxicity of COF-818 and COF-919 to the cells are both relatively low under dark conditions. Following 660 nm laser irradiation, the IC_{50} value of COF-818 and COF-919 were found to be $84.33 \mu\text{g mL}^{-1}$ and $42.29 \mu\text{g mL}^{-1}$, respectively. Similarly, upon 808 nm laser irradiation, the IC_{50} value of COF-818 and COF-919 were found to be $79.49 \mu\text{g mL}^{-1}$ and $67.78 \mu\text{g mL}^{-1}$, respectively. Finally, upon 660 + 808 nm laser irradiation, the IC_{50} value of COF-818 and COF-919 were found to be $62.95 \mu\text{g mL}^{-1}$ and $31.28 \mu\text{g mL}^{-1}$, respectively.

Figure R10. IC_{50} value of 4T1 cells treated with COF-818 and COF-919 at different concentrations under 660 nm and/or 808 nm laser irradiation.

Revision made:

We have added the data of Fig. R10 as “Supplementary Fig. S14” in the Supplementary Information and highlighted in BLUE on page 18.

Comment 4b:

4b. “(b) It is not mentioned in the manuscript what is the power and time of the laser used for the cell viability assay?”

Response 4b:

For cell viability assay, the 660 nm laser was set at a power of 0.5 W cm^{-2} , while the 808 nm laser was set at a power of 1.5 W cm^{-2} . The irradiation time was standardized at 5 min for both the 660 and 808 nm lasers.

Revision made:

We have revised the manuscript and highlighted in BLUE on page 17, with details as following:

4T1 cells were incubated overnight in RPMI-1640 medium, following which the culture

medium was substituted with varying concentrations (0, 10, 30, 50, 70, 90 $\mu\text{g mL}^{-1}$) of COF materials and irradiated under 660 nm (0.5 W cm^{-2} , 5 min) and/or 808 nm (1.5 W cm^{-2} , 5 min) lasers. After 24 h of incubation, the viability of 4T1 cells was assessed using CCK8 and MTT.

Comment 4c:

4c. “(c) What is the concentration of COFs used for the cell viability and the ROS generation assays?”

Response 4c:

To determine the impact of varying concentrations of COFs on cell viability, multiple concentrations (0, 10, 30, 50, 70, 90 $\mu\text{g mL}^{-1}$) were employed in the cell viability assay. For the ROS generation assays, a concentration of 100 $\mu\text{g mL}^{-1}$ COFs was utilized.

Comment 4d:

4d. “(d) Why nanoscale COF samples were generated by ultrasound treatment (Fig. 4j)?”

Response 4d:

Thanks for raising this important concern. In fact, the as-synthesized COF powder is prone to aggregation, resulting in the formation of large particles with microscale dimensions. Therefore, in order to achieve a well-dispersed COF nanoparticle, sonication was employed using an ultrasonic cell shredder.

Figure R1. Illustration of the strategy of disperse microsize COF sample to nanosize COF sample vis sonication.

Comment 4e:

4e. “(e) The authors should perform MTT assay to evaluate the cell viability in presence of laser and dark.”

Response 4e:

We thank this reviewer for the constructive suggestion. As the reviewer suggested, the viability of 4T1 cells treated with different conditions were evaluated by MTT assay, with details as following:

Figure R11. MTT assay of the viability of 4T1 cells treated with COF-818 and COF-919 at different concentrations under 660 nm and/or 808 nm laser irradiation.

Revision made:

We have added the data of Fig. R11 as “Supplementary Fig. S13” in the Supplementary Information and highlighted in **BLUE** on page 18.

Comment 4f:

4f. “(f) Control cells should be treated with only lasers to show that cells are not killed by the laser itself.”

Response 4f:

We thank this reviewer for the constructive suggestion. As the reviewer suggested, we conducted cell viability assays on 4T1 cells treated with PBS, PBS + 660 nm, PBS + 808 nm, and PBS + 660 + 808 nm laser to demonstrate that laser irradiation alone did not exert a significant impact on cell viability.

Figure R12. MTT assay of the viability of 4T1 cells treated with PBS, PBS + 660 nm, PBS + 808 nm, and PBS + 660 + 808 nm laser.

Revision made:

We have added the data of Fig. R12 as “Supplementary Fig. S15” in the Supplementary Information and highlighted in **BLUE** on page 19.

Comment 4g:

4g. “(g) What is the concentration of the COFs used for pyroptosis and ferroptosis assays?”

Response 4g:

A concentration of 100 $\mu\text{g mL}^{-1}$ COFs was employed for pyroptosis and ferroptosis assays.

Revision made:

We have revised the manuscript and highlighted in BLUE on page 17 to 18, with details as following:

Pyroptosis assay. 4T1 cells were incubated with COF samples ($100 \mu\text{g mL}^{-1}$) for 24 h. The resulting samples were subjected to different treatments and irradiated under 660 nm (0.5 W cm^{-2} , 5 min) and/or 808 nm (1.5 W cm^{-2} , 5 min) lasers. The cell supernatants were collected after 6 h, and the extracellular release of ATP was measured using an ATP assay kit (Beyotime Biotechnology, S0026). The experimental LDH release was determined using a LDH release kit (Beyotime Biotechnology, C0016).

Ferroptosis assay. 4T1 cells and COF samples ($100 \mu\text{g mL}^{-1}$) were incubated for 24h, then irradiated with 660 nm (0.5 W cm^{-2} , 5 min) and/or 808 nm (1.5 W cm^{-2} , 5 min) lasers. Then, the cells were used for GSH/GSSG assay. The ratio of GSH/GSSG in cells was measured using the GSH/GSSG assay kit (Beyotime Biotechnology, S0053) as per the manufacturer's instructions. Total GSH and oxidized GSH were quantified using a luminescence reader.

Comment 4h:

4h. “(h) What was the power and time of the laser irradiation used for the pyroptosis and ferroptosis assays?”

Response 4h:

For the pyroptosis and ferroptosis assays, the power of the 660 nm laser was set at 0.5 W cm^{-2} , while that of the 808 nm laser was set at 1.5 W cm^{-2} . The irradiation time was kept constant at 5 min for both the 660 and 808 nm lasers.

Revision made:

We have revised the manuscript and highlighted in BLUE on page 17 to 18, with details as following:

Pyroptosis assay. 4T1 cells were incubated with COF samples ($100 \mu\text{g mL}^{-1}$) for 24 h. The resulting samples were subjected to different treatments and irradiated under 660 nm (0.5 W cm^{-2} , 5 min) and/or 808 nm (1.5 W cm^{-2} , 5 min) lasers. The cell supernatants were collected after 6 h, and the extracellular release of ATP was measured using an ATP assay kit (Beyotime Biotechnology, S0026). The experimental LDH release was determined using a LDH release kit (Beyotime Biotechnology, C0016).

Ferroptosis assay. 4T1 cells and COF samples ($100 \mu\text{g mL}^{-1}$) were incubated for 24h, then irradiated with 660 nm (0.5 W cm^{-2} , 5 min) and/or 808 nm (1.5 W cm^{-2} , 5 min) lasers. Then, the cells were used for GSH/GSSG assay. The ratio of GSH/GSSG in cells was measured using the GSH/GSSG assay kit (Beyotime Biotechnology, S0053) as per the manufacturer's instructions. Total GSH and oxidized GSH were quantified using a luminescence reader.

Comment 4i:

4i. “(i) For the *in vivo* studies, what is the dose of the COFs used per mice and what was the power of the laser used for how long?”

Response 4i:

In the *in vivo* studies, 4T1 tumors were incubated with 100 μL of COF samples ($100 \mu\text{g mL}^{-1}$) per mouse. The 660 nm laser was set at a power of 0.5 W cm^{-2} , while the 808 nm laser was set

at a power of 1.5 W cm^{-2} . The irradiation time was standardized at 5 min for both the 660 and 808 nm lasers.

Revision made:

We have revised the manuscript and highlighted in BLUE on page 19, with details as following:

To establish a 4T1 mouse model, the right flank of BALB/c mouse was injected with 1×10^6 4T1 cells. The model was then randomly assigned to six groups, including PBS, 660 nm (0.5 W cm^{-2} , 5 min) plus 808 nm (1.5 W cm^{-2} , 5 min), COF-919, COF-919 + 660 nm (0.5 W cm^{-2} , 5 min), COF-919 + 808 nm (1.5 W cm^{-2} , 5 min), and COF-919 +660 nm (0.5 W cm^{-2} , 5 min) + 808 nm (1.5 W cm^{-2} , 5 min). The injected concentration of COF was $100 \mu\text{g mL}^{-1}$ ($100 \mu\text{L}$) per mouse.

Reviewers' Comments:

Reviewer #1:

Remarks to the Author:

This version is acceptable.

Reviewer #2:

Remarks to the Author:

The manuscript can be accepted

Reviewer #3:

Remarks to the Author:

The authors revised the manuscript as suggested and answered all the questions raised by the reviewer. This manuscript is recommended to be published in this current form.